# Contamination and Control of Mycotoxins in Grain and Oil Crops

**DOI:** 10.3390/microorganisms12030567

**Published:** 2024-03-12

**Authors:** Chenchen Zhang, Zheng Qu, Jie Hou, Yanpo Yao

**Affiliations:** 1Agro-Environmental Protection Institute, Ministry of Agriculture and Rural Affairs, Tianjin 300191, China; chynacici0314@163.com (C.Z.); quzhengwf@163.com (Z.Q.); jie_hou@yeah.net (J.H.); 2Xiangtan Experimental Station of Agro-Environmental Protection Institute, Ministry of Agriculture and Rural Affairs, Xiangtan 411199, China

**Keywords:** grain and oil crops, mycotoxin contamination, biological control

## Abstract

Mycotoxins are carcinogenic, teratogenic and mutagenic toxic compounds produced by some filamentous fungi, which are extremely harmful to corn, rice, wheat, peanut, soybean, rapeseed and other grain and oil crops, and seriously threaten environmental safety, food safety and human health. With the rapid increase in the global population and the expansion of the main crop planting area, mycotoxin contamination has increased year by year in agricultural products. The current review aimed to summarize the contamination status and harmful effects of major mycotoxins of grain and oil crops and the environmental factors that impact mycotoxin contamination. Further, control measures of mycotoxin contamination, especially the biological control strategies, were discussed.

## 1. Introduction

Mycotoxins are toxic secondary metabolites produced by certain filamentous fungi that pose a serious threat to human and animal health. Mycotoxin contamination results in waste of food and animal feed and impedes the international trade of crop products. According to the research of the Food and Agriculture Organization of the United Nations (FAO), global crop products exhibited a mycotoxin contamination rate that could reach as high as 25%, with approximately 2% resulting in complete loss of nutritional and economic properties [1]. Mycotoxin contamination could cause economic losses of tens of billions of dollars in agriculture and industry worldwide [2].

Most mycotoxins have the characteristics of nephrotoxicity, hepatotoxicity, carcinogenicity, teratogenicity, immune toxicity, neurotoxicity, mutagenicity and so on, thus seriously endangering the health of humans and animals [3,4,5]. As a series of low-molecular-weight compounds, mycotoxins can enter the human food chain directly through the ingestion of contaminated crop and crop products or indirectly through the consumption of animal products from livestock fed by contaminated feeds [6]. Mycotoxins can accumulate in mycotoxigenic fungi-infected maize, wheat, rice, rapeseed, soybeans, sorghum, peanuts and other grain and oil crops during the pre- and post-harvest stages. Mycotoxins are one of the main hazards responsible for increased “Rapid Alert System for Food and Feed” notifications and border rejections on crops and crop products exported to the European Union countries. Aflatoxins (AFs) are the most common mycotoxins. In 1960, the initial discovery of AFs began when a total of 100,000 turkeys died by turkey “X” disease in England. AFs are a significant global threat to human health, imposing not only a substantial economic loss in agriculture but also contributing to the millions of annual cases of cancer. Of all known AFs, aflatoxin B_1_ (AFB_1_) is the most toxic compound and is classified as a class I carcinogen by the International Agency for Research on Cancer, with toxicity more than 68 times greater than that of arsenic [7].

Grain and oil crops are pivotal agricultural commodities that significantly impact the national economy, people’s livelihoods, and overall food security. The quality and safety of these products can directly influence food safety and international food prices. Mycotoxin contamination is a major threat to the quality of grain and oil crop products, and therefore the prevention and control of mycotoxin contamination in grain and oil crops is a top priority for all major grain- and oil-producing countries. In China, the grain reserves held by farmers constitute approximately 50% of the nation’s overall grain production, which amounts to around 250 million tons. According to a sample survey of National Food and Strategic Reserves Administration of China, the average loss rate of stored grain is about 8% (40 million tons), with mycotoxin contamination accounting for the primary cause of this loss (about 30%) [8]. Given that mycotoxin contamination of grain and oil crops is serious and widespread, poses a huge threat to human health and food security and has been seriously neglected for a long time, further enhancement of the understanding and management of mycotoxin contamination has become an urgent issue. This paper reviews the current situation of mycotoxin contamination in grain and oil crops and its harmful effects and analyses the environmental factors affecting mycotoxin contamination. At the same time, due to the great potential shown by the biological control strategy in the management of mycotoxin contamination in crops, which provides a safe and effective alternative to chemical fungicides, the article also proposes a biological control strategy for mycotoxin contamination in grain and oil crops in conjunction with the progress of related research, with the aim of providing theoretical references to the in-depth understanding and scientific prevention and control of mycotoxin contamination in grain and oil crops.

## 2. Methodology

PubMed, Web of Science, Google Scholar and Science Direct databases were used to search literature published from 1980 onwards featuring keywords and signatures in this study. Updates from the World Health Organization (WHO) and other governmental organizations around the world (Food and Agriculture Organization of the United Nations (FAO), European Food Safety Authority (EFSA), and the Food and Drug Administration (FDA)) were also reviewed. The keywords used were: “mycotoxins”, “grain and oil crops”, “contamination”, “harmful effects”, “environmental factors”, “prevention and control measures” and “biological control”. In addition, a supplementary literature search was conducted using research bibliographies relevant to the purpose of this study. After this selection, the full text was analyzed and summarized, and articles that did not meet the purpose of this review were excluded, mainly for the following reasons: (i) research on mycotoxin contamination of crops other than grain and oil crops; (ii) unfinished/unpublished research; (iii) abstracts of conferences.

## 3. Status and Harmful Effects of Mycotoxin Contamination in Grain and Oil Crops

Mycotoxins can harm plants and animals, and at the same time, mycotoxin contamination has a wide range of toxicity, and mycotoxins, once they enter the food chain, can have a huge impact on human health and socio-economics. At the same time, due to the wide global distribution of fungi, their products, mycotoxins, have relatively stable physicochemical properties. In general, all crops can be contaminated by mycotoxins, and mycotoxins are more widely found in grain and oil crop products. Of the many mycotoxins, only a few regularly contaminate agricultural products and animal feed, such as aflatoxin (AFs), ochratoxin, patulin, zearalenone (ZEA), fumonisin (FUM), deoxynivalenol (DON) and T-2 toxin (T-2) [9,10,11,12,13]. Mycotoxins are not essential for the growth or development of mycotoxigenic fungi; however, they significantly influence the pathogenicity, aggressiveness, and virulence of mycotoxigenic fungi. There are three major toxigenic fungal genera: *Aspergillus*, *Fusarium* and *Penicillium*. Usually, aflatoxin, ochratoxin and patulin are produced by *Aspergillus* and *Penicillium* fungi; ZEA, FUM, DON and T-2 are mainly produced by the *F. graminearum*, *F. moniliforme*, *F. sporotrichioides*, *F. tricinctum* and other *Fusarium* fungi [14,15,16,17] (Figure 1).

Maize is one of the most widely grown crops in the world and is considered to be one of the most susceptible to mycotoxins, with the toxin species that have the most contaminating effects on maize including ZEA, DON and AFB_1_ [18]. As the top-producing food crop in the world, there are more than 170 maize-growing countries and regions. From 1967 to 2019, the total yield of maize increased from 272 million to 1.11 billion tons [19]. Globally, about 25% of maize and its products are contaminated with mycotoxins. This is a worldwide problem for food safety and public health [20]. Among them, AFs are one of the most important mycotoxins in agriculture. *Aspergillus flavus* is known to be the main AF-producing fungi, and its suitable temperature is also close to the optimal growth temperature of common crops. China’s annual loss of food due to AFs contamination amounted to more than 20 billion kilograms. According to the abundance in contents of calcium, carbohydrates, fatty acids, dietary fiber, phosphorus, proteins and vitamins, peanut, the world’s fourth-largest oilseed crop, has high nutritional and commercial value [21]. Peanut is one of the most susceptible crops for *A. flavus* infection [22,23]. In China, where annual peanut production accounts for more than 42% of the world’s peanut production, *A. flavus* infection could cause serious losses [24].

The distributions of mycotoxigenic fungi depended on geographical regions, cultivation methods, conditions of processing, transportation and storage of food and oil crops. The production of mycotoxins is the result of the combined action of mycotoxigenic fungi and environmental factors. Mycotoxin contamination can occur throughout the entire chain of production for grain and oil crops, including growing, harvesting, storage, transportation, and processing, as long as the conditions are suitable, and simultaneous contamination with multiple mycotoxins can occur, while synergistic interactions between the toxins can lead to the accumulation of toxins and enhancement of their toxic effects. Environmental conditions are key factors influencing fungal colonization and mycotoxin production. The prevention and control of mycotoxins has always been a problem for all countries in the world. Mycotoxin contamination has the characteristics of strong concealment, a long latency period and the occurrence of multiple toxins simultaneously. If the grain seeds do not show obvious signs of mold or other diseased symptoms, it is difficult to visually identify the presence of fungal toxins.

## 4. Environmental Factors of Mycotoxin Contamination in Grain and Oil Crops

Climate is a key driver of mycotoxigenic fungi colonization and mycotoxin production. The complex and ever-changing climate, and especially the occurrence of more and more frequent extreme climate events, always threaten the global food and feed supplies. It is therefore extremely important to focus on the impact of climate change (CC) on mycotoxin production [25]. Stricter control of mycotoxins is a natural trend worldwide due to their complex toxicities and widespread distribution in food and feed globally. However, the changing global environment adds significant difficulties to the control and reduction of mycotoxin exposure. Mycotoxin production is very sensitive to environmental factors. Therefore, the communities of toxin-producing fungi will be affected when the weather changes. More complexly, it might alter the dominance of other microorganisms in the drying and storage stages of grain and oil crops, resulting in mycotoxin contamination during transportation and storage stages [26]. Maize and peanuts are particularly susceptible to infection by mycotoxigenic fungi during periods of drought. In developing countries, drought stress might be particularly important in terms of food security. For example, in West and East Africa, where marginal lands previously planted with sorghum have now been replaced by stress-tolerant maize, the production of mycotoxins in the field is affected by high temperatures and water stress, which makes maize and groundnuts particularly susceptible because of the increased adaptability of mycotoxins to the tropics and sub-tropics when a period of water stress arrives. Crops are also very susceptible to mycotoxin contamination during post-harvest storage, where temperature and relative humidity are important factors when storing grain and oil crops. This is because these factors affect the equilibrium moisture content of the grain, which is essential for controlling the growth of toxin-producing fungi during storage [27,28]. These led to an increase in mycotoxin contamination of such crops pre- or post-harvest periods, severely affecting the quality and economic value of those crops [29]. In this section, we summarized the interaction of environmental factors affecting the production of mycotoxins in grain and oil crops and analyzed the influence of these interacting environmental factors on mycotoxin contamination.

### 4.1. Interaction between Environmental Factors Affecting Mycotoxin Contamination of Grain and Oil Crops

The relationship between CC and food security has attracted much attention worldwide in the last decade, leading to concerns about the effects of the interaction of climate-related factors, such as high temperatures, drought stress and elevated CO_2_ concentrations [CO_2_], on mycotoxigenic fungi pathogenicity and mycotoxin contamination of major food crops [30,31] (Figure 2). Interacting environmental factors, particularly temperature and water activity (a_w_), are critical in mycotoxigenic fungi colonization and mycotoxin contamination. For example, in the case of AFs, the production of AFs usually increases at relatively high a_w_, i.e., there is a positive correlation between the two, while in the case of temperature, the optimum temperature for AFs production is between 24 °C and 30 °C, but there is some variation depending on the strain and substrate. Recent studies have shown that environmental factors such as a_w_, temperature, [CO_2_] and light all have a significant effect on growth, development and mycotoxin production of mycotoxigenic fungi [32,33,34,35]. A great deal of research suggests that the interactions of a_w_ and temperature were the key factors regulating the growth and secondary metabolite production of mycotoxigenic fungi, but the levels of [CO_2_] altered the responses of mycotoxigenic fungi to changes in water and temperature [36]. Specifically, we should fully consider the influences of drought stress, global warming and raised [CO_2_], which could impact the mycotoxin contamination of grain and oil crops during pre- or post-harvest stages. It was predicted that the global average temperature would increase by 4 °C and [CO_2_] would reach approximately 1000 ppm in 2100 [37]. In addition, rainfall patterns would be expected to change. Human industrial activities would lead to the frequent occurrence of droughts and flooding in a cyclical pattern [38,39]. Moreover, other factors, such as ultraviolet light and diurnal temperature range, could also affect mycotoxigenic fungi infection and mycotoxin contamination [29,40,41]. Ongoing environmental changes are slowly but steadily shaping the relationship between plant and fungal pathogens [42,43,44]. In the coming decades, the potential impacts of CC on food security and food safety will become even more important.

### 4.2. Influence of Interacting Environmental Factors on Mycotoxin Contamination of Grain and Oil Crops

Currently, there are still limited studies and predictions regarding the impact of climate change on plant fungi diseases and mycotoxin contamination. The existing research is mainly based on the historical or current climatic data and the results of interaction between a_w_ and temperature [45]. Relatively few studies have studied the tripartite interactions between temperature, a_w_ and [CO_2_] and the physiological and ecological changes that may occur during the accumulation of toxin-producing fungi and mycotoxins. Recently, researchers have taken a keen interest in the adaptation of mycotoxigenic fungi, including *Aspergillus* and *Fusarium*, under those environmental factors. For example, a study showed that under conditions of high temperature, low a_w_ and elevated [CO_2_], the growth of *A. flavus* on corn kernels was normal, but the expression of AFs biosynthesis genes was up-regulated and AFs production was increased [46]. Under conditions of high temperature and elevated [CO_2_], the infection ability of *F. verticillioides* on maize increased, but the level of FUM was not. However, when drought stress was introduced into the environmental conditions, the production of FUM increased [47,48,49]. Studies of ochratoxin A contamination of stored coffee produced by *A. ochratoxin* and *A. carbonarius* suggest that there may be some differences in the effects of these interacting climate change-related environmental factors on different toxin-producing fungi. For example, under environmental conditions of high temperature and moisture stress, there is an inhibitory effect on *A. ochratoxin* toxicity production, whereas there is no effect on *A. carbonarius* toxicity production [50]. The severity of Septoria tritici blotch (STB) and Fusarium head blight (FHB) of wheat increased as the [CO_2_] raised from 390 ppm to 780 ppm, with a more pronounced impact on FHB compared to STB [51].

CC might favor the colonization of thermophilic fungi, as they can grow under very hot and dry conditions [43,52,53]. The secondary metabolites of thermophilic fungi might play a key role in intraspecific competition [54]. For example, the thermophilic fungi, *Wallemia sebi*, could produce metabolites toxic to humans and animals [55]. This phenomenon has a significant impact on agricultural productivity, especially for major grain and oil crops, such as maize and peanuts. It might also affect the interaction between crops, fungi and insects. This would have far-reaching implications on the issue of mycotoxin contamination in grain and oil crops, especially AFs contamination in developing countries [56,57].

## 5. The Control Strategies for Mycotoxin Contamination in Grain and Oil Crops

The best practice for food crops with excessive mycotoxin contamination is to discard them, but this not only causes huge losses for the economy, but also creates a serious environmental contamination problem that continues to pass mycotoxins through the food chain. Therefore, there is an urgent need for research to discover efficient, safe, healthy and easy-to-use measures for controlling mycotoxin contamination. There were several control strategies for mycotoxin contamination in grain and oil crops, including development of biological control agents (BCAs), breeding of resistant varieties, use of chemical fungicides, rational irrigation, insect control, harvesting at the right time, avoiding mechanical damage during harvest stage, rapid drying after harvest and controlling the temperatures and relative humidity during storage [58,59,60] (Figure 3). The mechanism of mycotoxin biological control involves competition between one or several BCAs and mycotoxigenic fungi for nutrients (e.g., organic humus, nitrogen, phosphorus, potassium, nitrite, iron and other trace elements), ecological niches, water, and air, thereby inhibiting the growth, pathogenicity and reproduction of mycotoxigenic fungi [61]. Among them, the use of BCAs has the great potential in controlling and reducing mycotoxin contamination of crops and is widely considered as a good alternative to chemical fungicides. Mycotoxin contamination can occur at all stages of crop growth; therefore, the optimal approach to reducing mycotoxin exposure and its related health hazards is to combine the control measures during pre- and post-harvest in order to ensure safe crop production and human health [62,63]. In this section, we outline the main biological control strategies for mycotoxin contamination management and indicate the standards for controlling mycotoxin levels in food in different countries and regions.

### 5.1. The Control of Mycotoxin Production

Mycotoxigenic fungi are ubiquitous in the environment, especially in soil and crop residues, and timely measures are needed to control the production of mycotoxins. In the soil, mycotoxigenic fungi participate in the cycling of organic matter by decomposing plant material. The conidia of mycotoxigenic fungi can be dispersed by wind and vector insects, allowing them to infect new host plants. In maize, mycotoxigenic fungi firstly infect the corn silk and then grow down along the corn silk and infect developing kernels, ultimately producing mycotoxins [64]. The infection of mycotoxigenic fungi in the field is the beginning of the mycotoxin contamination process. Therefore, comprehensive management of mycotoxin should start in the field before harvest.

#### 5.1.1. Biological Control of Antagonistic Microorganisms

There are many antagonistic microbes that can effectively control the infection of mycotoxigenic fungi, such as *Bacillus*, *Lactobacillus*, *Pseudomonas*, *Klebsiella*, and *Burkholderia* [65]. As a kind of distinguished representative mycoparasitism fungi, *Trichoderma* has a significant inhibitory effect on a variety of mycotoxigenic fungi, such as *F. graminearum*, *F. cucumber*, and *F. oxysporum* [66,67,68].

#### 5.1.2. Biological Control of Non-Toxigenic Fungal Strains

The non-toxigenic fungal strain is a specific type of fungi that inherently lacks the capacity to synthesize mycotoxins, thereby exhibiting potential as an effective BCA to manage mycotoxin contamination. The non-toxigenic *A. flavus* is not only a good competitor against mycotoxigenic *A. flavus*, but also can activate the defense response of the host plant. Moreover, some non-toxigenic *A. flavus* can secrete secondary metabolites to inhabit the AFs production in mycotoxigenic *A. flavus* [69,70,71]. Treatment with non-toxigenic *A. flavus* could cause a significant and sustained reduction in AFs of 70–90% in peanut and cotton [72]. Now, a series of soil treatment agents with non-toxigenic strains has been developed. Soil treatment with competitive non-toxigenic strains has been shown to have a sustained effect that protects grain and oil crops from contamination by toxin-producing fungi during storage [73]. However, the most effective method is seed coating agents. After treated with seed coating agents, the conidia of non-toxigenic strains could cover the entire seed. The abundant spores provide a sufficient inoculum to achieve the competitive advantage of non-toxic strains.

#### 5.1.3. Biological Control of Mixed Strains

Generally, the BCAs used to control mycotoxin production in grain and oil crops are of a single strain. However, Probst et al. [74] concluded that mixed strains were more able to compete effectively than single strains in complex environments.

The colonization and growth of BCAs are influenced by environmental factors, particularly humidity and temperature [33]. Therefore, when using biological control agents in the field, it is important to consider their adaptability to environmental conditions. In addition, targeted and timed application of BCAs is a critical step in successfully controlling the growth of mycotoxigenic fungi and inhibiting mycotoxin contamination.

### 5.2. Bio-Detoxification of Mycotoxin

Unlike in the mycotoxin production stage through antagonists and other inhibitors of mycotoxin production by mycotoxigenic fungi, and thus the control of crop mycotoxin, if mycotoxin contamination of the crop has already occurred, certain detoxification measures should be taken. Traditional physical and chemical methods for the removal of mycotoxins have certain limitations, resulting in serious crop nutrient loss, making them more difficult to use in production practice. In contrast, the biological detoxification of mycotoxins (biosorption and biodegradation) offers a more environmentally friendly and convenient alternative to physical and chemical approaches.

#### 5.2.1. Biosorption of Mycotoxins

Biosorption is a physico-chemical process and is reversible. In this process, mycotoxins can be adsorbed into the cell walls of microorganisms, and even dead cells still have adsorption capacity [75]. A variety of microorganisms have been found to have the ability to bind and adsorb AFB_1_, among which yeast and lactic acid bacteria have been widely studied due to their high adsorption capacity and high food safety. It has been suggested that AFB_1_ is able to bind to the surface components of the cell wall of *Lactobacillus*, and this binding is reversible and the bonding strength depends on the probiotic affinity toward AFs, the environment and the treatment conditions, whereas the adsorptive binding of yeast to AFB_1_ occurs via the mannan of its cell wall [76,77]. The above interactions between yeast and mycotoxins as well as between lactic acid bacteria and mycotoxins show that the structural integrity of the cell wall, the physical morphology structure as well as the chemical composition play important roles in the adsorption of mycotoxins.

#### 5.2.2. Biodegradation of Mycotoxins

Biodegradation is a very promising and environmentally friendly method for the degradation of mycotoxins, and applications of this method include the use of microorganisms or the use of enzymes. Among the methods of microbial degradation of mycotoxins are the use of microbial catabolic pathways to detoxify mycotoxins into less toxic intermediates or end products. This method is relatively gentle and does not damage the quality of agricultural products, but also enhances the nutritional value of agricultural products, so the degradation of mycotoxins by microorganisms has attracted widespread attention around the world. A variety of bacteria and fungi with biodegradation capabilities have been identified. These bacterial species include *Nocardia corynebacteroides*, *Corynebacterium rubrum* and *Rhodococcus* spp. [78]. *L. plantarum* is considered to be the most effective microorganism for degrading AFB_1_ [79]; *P. aeruginosa* exhibited degradation rates of 82.8%, 46.8% and 31.9% for AFB_1_, AFB_2_ and AFM_1_, respectively, when incubated in a medium at 37 °C for 76 h [80]. Many enzymes from microorganisms have been reported to degrade mycotoxins both in vivo and in vitro, including oxidases, peroxidases, laccases, reductases, esterases, carboxylesterases, transaminases and lactose hydrolases [81]. In contrast to the application of microorganisms for mycotoxin degradation, the application of enzymes ensures homogeneity and reproducibility, and is safe and easy to handle without the risk of contamination. It was found that extracellular extracts of *Rhodococcus erythropolis* could be effective in degrading AFs [82]. By optimizing the medium and optimal fermentation conditions for enzyme production of *Myxococcus fulvus* with good mycotoxin reduction effect, the degradation rate of AFB_1_ by *M. fulvus* could reach 80.7% [83]. Further studies showed that the extracellular metabolites of *M. fulvus* ANSM068 could not only degrade AFB_1_, but also had high degradation activity for AFG_1_ and AFM_1_ [84].

### 5.3. Regulatory Mechanisms of Mycotoxins

In addition to controlling mycotoxin contamination through biological control strategies of mycotoxins, in order to ensure food safety, countries and international organizations usually set limits for mycotoxins in food. Currently, a total of more than 400 mycotoxins have been identified, and more than 100 countries and organizations have set guidance levels of major mycotoxins in food and feed [85]. As can be seen from the table, the levels of mycotoxins in food and oil crops are strictly regulated and standardized in many countries and regions around the world. The Joint Food and Agriculture Organization of the United Nations (FAO)/World Health Organization (WHO) Expert Committee on Food Additives (JECFA) recommendations for the provisional maximum tolerable daily intake (PMTDI) are 2 µg/kg bw/d each for Fumonisin B_1_ (FB_1_), Fumonisin B_2_ (FB_2_) and Fumonisin B_2_ (FB_2_), 0.5 µg/kg bw/d for zearalenone (ZEA), 0.025 µg/kg bw/d each for T-2 toxin (T-2) and HT-2 toxin (HT-2), and 1 µg/kg bw/d for deoxynivalenol (DON). The tolerable daily intake (TDI) is also established by the European Food Safety Authority (EFSA), which is 1 µg/kg bw/d for each of FB_1_, FB_2_ and FB3, 0.25 µg/kg bw/d for ZEA, 0.02 µg/kg bw/d each for T-2 and HT-2, 1 µg/kg bw/d for DON and 1 µg/kg bw/d for nivalenol (NIV). In addition, the FAO, China, the European Union (EU) and the USA Food and Drug Administration (FDA) have established maximum intake levels for different grain crops and their related products, such as barley, wheat, maize, wheat flour, maize flour and so on (Table 1).

## 6. Conclusions

Grain and oil crops and their products are the most important agricultural products that have a bearing on the national economy and people’s livelihoods. The quality and safety of these crops have a direct impact on the basic supply of national food, food security, and international food prices. Consequently, they have traditionally been the focus of attention of countries around the world. However, with the frequent occurrence of climate and environmental problems such as global warming and extreme droughts in recent years, the mycotoxin contamination of grain and oil crops products has been worsening year by year. Mycotoxins can accumulate in grain and oil crops during all pre-harvest and post-harvest stages. Therefore, the control of mycotoxin in crops and products has emerged as a significant issue. At present, the management of mycotoxin contamination in grain and oil crop products has been carried out in a number of countries around the world, but the effects have not been satisfactory. With the development and advancement of biotechnology, agricultural scientists are considering using beneficial microorganisms to control mycotoxin contamination. The use of beneficial microorganisms to control and reduce mycotoxin contamination of food and oil crops is very effective and has broad application potential. It is widely considered a safe and efficient alternative to chemical fungicides for controlling mycotoxin contamination. In order to harness the full potential of microorganisms, apart from considering environmental factors, it is also necessary to explore the interactions between microorganisms and build the beneficial artificial microbial community. Mycotoxin contamination can occur at all stages of grain and oil crops. Therefore, the best approach is to combine pre-harvest and post-harvest control measures in an integrated management approach that can provide for the safe production of food and oil crops as well as human health.

Regulation of mycotoxins can also be achieved by functional microorganisms. For example, by applying functional microorganisms to the soil, these beneficial strains interfere with the proliferation of native toxin-producing fungi, enhance host resistance, control the growth of toxin-producing fungi before and after harvest and utilize their own natural metabolites to prevent mycotoxin production. The production of fungal toxins and their regulatory genes in the field is affected by high temperature, high humidity and drought stress, and the biosynthetic genes controlling the production of fungal toxins are usually clustered together, so the production of fungal toxins can be reduced or inhibited by inhibiting specific regulatory genes or structures, and drought-resistant, stress-resistant functional microorganisms can be used to achieve these objectives. Of course, for functional microorganisms to play a greater role, in addition to considering these environmental factors, it is also necessary to consider a combination of more powerful and mechanistically diverse synthetic microbiomes, which are key measures for the successful control of mycotoxin contamination in grain and oil crops and their products.

## Figures and Tables

**Figure 1 microorganisms-12-00567-f001:**
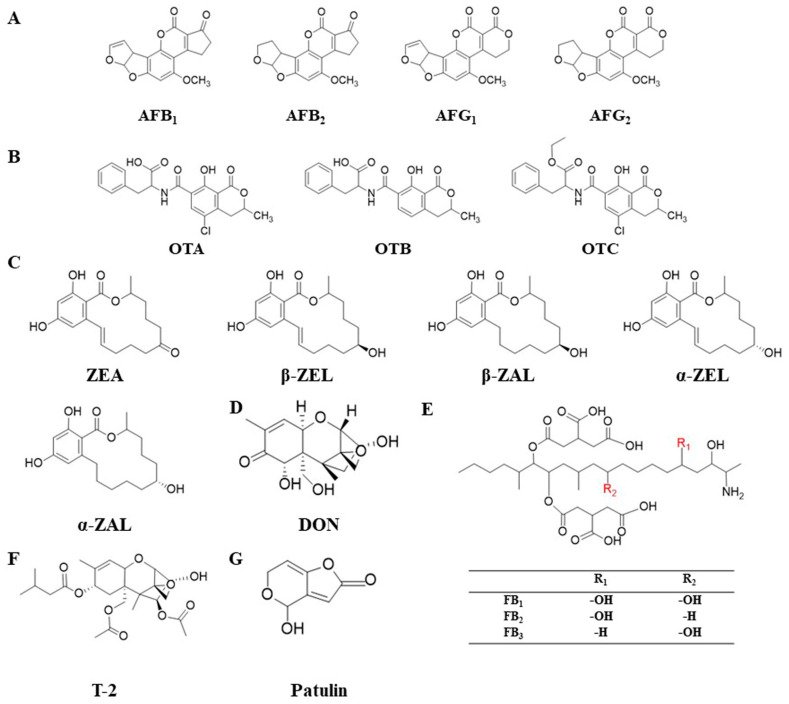
Chemical structure of common mycotoxins. (**A**) Chemical structure of aflatoxin. AFB_1_: aflatoxin B_1_; AFB_2_: aflatoxin B_2_; AFG_1_: aflatoxin G_1_; AFG_2_: aflatoxin G2. (**B**) Chemical structure of ochratoxin. OTA: ochratoxin A; OTB: ochratoxin B; OTC: ochratoxin C. (**C**) Chemical structure of zearalenone and its derivatives. ZEA: zearalenone; β-ZEL: β-zearalenol; β-ZAL: β-zearalanol; α-ZEL: α-zearalenol; α-ZAL: α-zearalanol. (**D**) Chemical structure of deoxynivalenol. DON: deoxynivalenol. (**E**) Chemical structure of fumonisins; FB_1_: Fumonisin B_1_; FB_2_: Fumonisin B_2_; FB_3_: Fumonisin B_3_. (**F**) Chemical structure of T-2 toxin. T-2: T-2 toxin. (**G**) Chemical structure of patulin.

**Figure 2 microorganisms-12-00567-f002:**
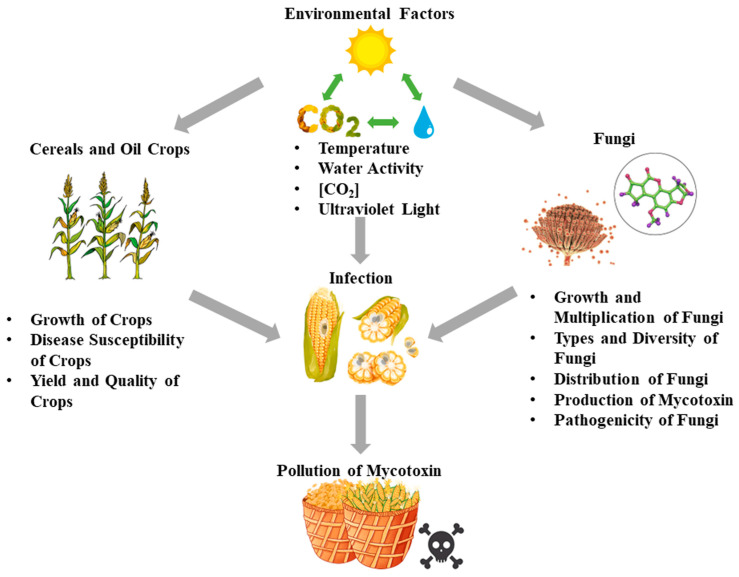
Impact of environmental factors on toxin contamination.

**Figure 3 microorganisms-12-00567-f003:**
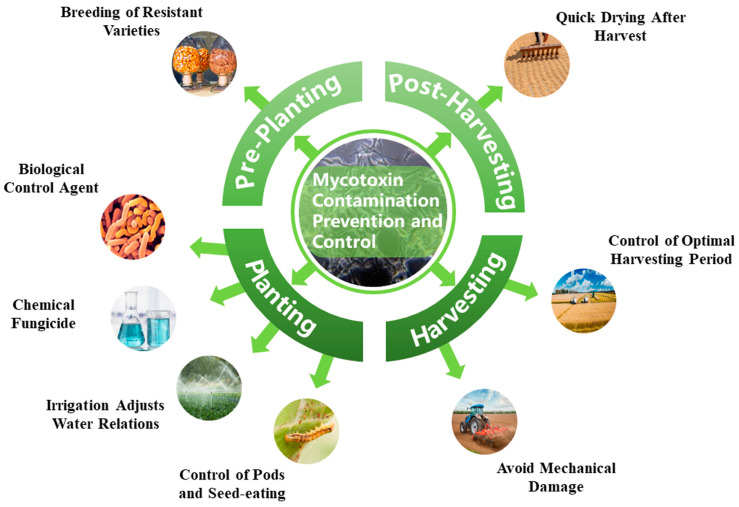
Control methods of mycotoxin contamination in grain and oil crops.

**Table 1 microorganisms-12-00567-t001:** The representative regulations of different organizations or countries on various mycotoxins.

Mycotoxin	PMTDI (JECFA)	TDI (EFSA)	CXS 193-1995 [86] (FAO)	GB 2761-2017 [87] (China)	Commission Regulation (EC) No 1881/2006 (EU) [85]	FDA (The USA)
AFB_1_				20 µg/kg (maize, maize flour, maize products) [87] 5 µg/kg (barley, wheat, cereal, wheat flour) [87]	2 µg/kg (all cereals and all products derived from cereals) [85]	
AFB_1_ + AFB_2_ + AFG_1_ + AFG_2_					4 µg/kg (all cereals and all products derived from cereals) [85]	
FB_1_, FB_2_, FB_3_	2 µg/kg bw/d [88]	1 µg/kg bw/d [89]				
FB_1_ + FB_2_			2000 µg/kg (maize flour and maize meal) [86] 4000 µg/kg (unprocessed maize kernels) [86]		4000 µg/kg (unprocessed maize) [85] 1000 µg/kg (maize for direct human consumption, and maize products) [85]	
FB_1_ + FB_2_ + FB_3_						Guideline: 2 ppm (maize products with germ removed) [90]
ZEA	0.5 µg/kg bw/d [91]	0.25 µg/kg bw/d [92]		60 µg/kg (wheat, wheat flour) [87] 60 µg/kg (maize, maize flour) [87]	100 µg/kg (unprocessed grains other than maize) [85] 75 µg/kg (cereals for direct human consumption, or grain flour, bran, embryo) [85]	
T-2, HT-2	0.025 µg/kg bw/d [93]	0.02 µg/kg bw/d [94]				
DON	1 µg/kg bw/d [95]	1 µg/kg bw/d [96]	1000 µg/kg (flour, semolina, semolina and flakes from wheat, maize or barley) [86] 2000 µg/kg (wheat, maize and barley for further processing) [86]	1000 µg/kg (barley, wheat, cereal, wheat flour) [87] 1000 µg/kg (maize, maize flour) [87]	1750 µg/kg (unprocessed durum wheat and barley) [85] 750 µg/kg (cereals for direct human consumption, or grain flour, bran, embryo) [85]	Recommended Standard: 1 ppm (cereals for direct human consumption, or grain flour, bran, embryo) [91]
NIV		1.2 µg/kg bw/d [97]				

PMTDI: provisional maximum tolerable daily intake; JECFA: Joint FAO/WHO Expert Committee on Food Additives; TDI: tolerable daily intake; EFSA: European Food Safety Authority; FAO: Food and Agriculture Organization of the United Nations; FDA: Food and Drug Administration; HT-2: HT-2 toxin; NIV: nivalenol.

## Data Availability

Not applicable.

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
