# Peer review of "Contamination and Control of Mycotoxins in Grain and Oil Crops"

_microorganisms, 2024, doi:10.3390/microorganisms12030567_

Round 1
Reviewer 1 Report
Comments and Suggestions for Authors
The paper is a review of mycotoxin contamination of grains, as well as the factors that facilitate contamination and the control strategies that are currently employed or have been proposed in various studies. The topic is important from a variety of economic, health and scientific perspectives. However, some inconsistencies and omissions should be resolved before accepting the final version.
The title of the manuscript is very general (The Mycotoxin Contamination in Grain and Oil Crops) and does not properly describe the content of the paper. The title suggests that there will be an analysis of mycotoxin contamination of food grains and oil crops. However, the manuscript emphasizes the establishment of maximum permissible limits for food and animal feed contamination, the impact of climate change and biological control, among other topics. It is suggested to adapt the title to the content of the study or vice versa.
The abstract indicates that the manuscript deals with a review of the harmful effects, environmental factors and control strategies, especially biological control, of major mycotoxin contamination of cereal and oilseed crops. However, the organization of the manuscript does not correspond to this description. There is no section on harmful effects, but these are mentioned very briefly in the introduction and in the following sections. It is suggested to adapt the abstract to the content of the manuscript or vice versa.
Similarly, the introduction does not present or explain the organization of the manuscript. Rather, the introduction is a thematic section of the manuscript that sets out, among other topics, the maximum permissible limits established in many countries and organizations. However, this information would correspond to contamination control strategies for the main mycotoxins. It is suggested to reorganize the introduction of the manuscript according to the usual style of a scientific review; it is suggested to present the problem as a framework to make explicit the intended approach of the study, the importance of the approach, the objective or scientific questions to be solved by the study, as well as to present the organization of the manuscript.
The methodology followed to select the bibliographic material that supports the study is not presented, so it is not possible to evaluate the comprehensiveness, precision or scope of the contents. It is suggested to briefly outline the steps followed to compile, analyze and organize the information presented.
The manuscript contains three thematic sections, in addition to the introduction and conclusions. These sections do not contain an organization or structure that makes them easy to read and understand. It is suggested to establish at the beginning of each section a brief presentation of the logical structure of the contents, which will be followed by the breakdown of each of the subsections.
Each of the subsections should contain absolutely all the information related to the topic. For example, information is detected that aflatoxin B1 is classified as a Class I carcinogen by IARC (Lines 40-43). Subsequently, (Lines 113-114) it is indicated that in 1993 the WHO classified aflatoxins as Class I carcinogens. Also, all types of grains and oilseeds that are contaminated are repeated several times throughout the manuscript (Lines 35-36, 85-86, 117-118, 338-339). Similarly, the historical information on turkey disease X (Lines 86-88) corresponds to an introductory section, rather than this section which should only include a thorough explanation of the biological and abiotic processes by which grains and oilseeds are contaminated by mycotoxins.
It is also suggested to ensure that each section contains all the complete information on the subject. For example, Table 1 on regulations on various mycotoxins omits regulations on aflatoxins, which contradicts several statements in the manuscript that point to these mycotoxins as the most common and dangerous, in addition to the fact that these regulations exist in different countries. In the same way, Figure 2 deduces from climate change absolutely all the environmental impact on contamination by these toxins, when environmental factors persistently act and influence contamination with and without this climate change. Likewise, in Figure 3 the absence of post-harvest control measures is notorious, which also contradicts several statements in the manuscript that point to fungi as dangerous in the post-harvest stage and storage of grains and oilseeds.
Comments on the Quality of English LanguageThe manuscript contains several inappropriate styles that we suggest correcting to improve the readability and comprehension of the manuscript. For example, the abbreviation for "aflatoxins (AFs)" is identified (line 39), but later these same mycotoxins are referred to as "AF" (lines 39, 41, 89, 112, etc.). Table 1 contains multiple abbreviations that are not defined in the table itself so one must resort to the text. The provisional maximum tolerable daily intake is defined in two ways (PTMDI and PMTDI) (lines 50 and 60). In the paragraph of lines 62-66 the simile of the Sword of Damocles is made with the contamination of grains and oilseeds by mycotoxins; which is an expression of vague meaning because each reader can interpret it differently according to his knowledge of classical Greek literature.
Author Response
Point-by-Point Responses to comments
Reviewer 1:
The paper is a review of mycotoxin contamination of grains, as well as the factors that facilitate contamination and the control strategies that are currently employed or have been proposed in various studies. The topic is important from a variety of economic, health and scientific perspectives. However, some inconsistencies and omissions should be resolved before accepting the final version.
Comments #1: The title of the manuscript is very general (The Mycotoxin Contamination in Grain and Oil Crops) and does not properly describe the content of the paper. The title suggests that there will be an analysis of mycotoxin contamination of food grains and oil crops. However, the manuscript emphasizes the establishment of maximum permissible limits for food and animal feed contamination, the impact of climate change and biological control, among other topics. It is suggested to adapt the title to the content of the study or vice versa.
Response: We have revised the title of the manuscript to " Contamination and Control of Mycotoxin in Grain and Oil Crops".
Comments #2: The abstract indicates that the manuscript deals with a review of the harmful effects, environmental factors and control strategies, especially biological control, of major mycotoxin contamination of cereal and oilseed crops. However, the organization of the manuscript does not correspond to this description. There is no section on harmful effects, but these are mentioned very briefly in the introduction and in the following sections. It is suggested to adapt the abstract to the content of the manuscript or vice versa.
Response: We have revised the original section of the manuscript, "The Mycotoxin Contamination in Grain and Oil Crops", to read "Status and Harmful Effects of Mycotoxin Contamination in Grain and Oil Crops".
Comments #3: Similarly, the introduction does not present or explain the organization of the manuscript. Rather, the introduction is a thematic section of the manuscript that sets out, among other topics, the maximum permissible limits established in many countries and organizations. However, this information would correspond to contamination control strategies for the main mycotoxins. It is suggested to reorganize the introduction of the manuscript according to the usual style of a scientific review; it is suggested to present the problem as a framework to make explicit the intended approach of the study, the importance of the approach, the objective or scientific questions to be solved by the study, as well as to present the organization of the manuscript.
Response: We have described the manuscript organization of the introduction and moved the national and regional limits for mycotoxins to the "The Control Strategies for Mycotoxin Contamination of Grain and Oil Crops" section under the heading "Regulatory Mechanisms for Mycotoxins".
Comments #4: The methodology followed to select the bibliographic material that supports the study is not presented, so it is not possible to evaluate the comprehensiveness, precision or scope of the contents. It is suggested to briefly outline the steps followed to compile, analyze and organize the information presented.
Response: We added a methodology section to the manuscript, please see lines 99-111.
Comments #5: The manuscript contains three thematic sections, in addition to the introduction and conclusions. These sections do not contain an organization or structure that makes them easy to read and understand. It is suggested to establish at the beginning of each section a brief presentation of the logical structure of the contents, which will be followed by the breakdown of each of the subsections.
Response: We have broken down the section of the article and added the section title. The fourth part of the article, "4 Environmental Factors of Mycotoxin Contamination in Grain and Oil Crops", is subdivided into two sections: "4.1 Interaction between Environmental Factors Affecting Mycotoxin Contamination of Grain and Oil Crops" and "4.2 Influence of Interacting Environmental Factors on Mycotoxin Contamination of Grain and Oil Crops". Similarly, the first section of "5.1 The Control of Mycotoxin Production" in the fifth part of "5 The Control Strategies for Mycotoxin Contamination in Grain and Oil Crops" is subdivided into three aspects: "5.1.1 Biological Control of Antagonistic Microorganisms", "5.1.2 Biological Control of Non-Toxigenic Fungal Strains" and "5.1.3 Biological Control of Mixed Strains". The second section "5.2 Bio-detoxification of Mycotoxin" is subdivided into "5.2.1 Biosorption of Mycotoxins" and "5.2.2 Biodegradation of Mycotoxins".
Comments #6: Each of the subsections should contain absolutely all the information related to the topic. For example, information is detected that aflatoxin B1 is classified as a Class I carcinogen by IARC (Lines 40-43). Subsequently, (Lines 113-114) it is indicated that in 1993 the WHO classified aflatoxins as Class I carcinogens. Also, all types of grains and oilseeds that are contaminated are repeated several times throughout the manuscript (Lines 35-36, 85-86, 117-118, 338-339). Similarly, the historical information on turkey disease X (Lines 86-88) corresponds to an introductory section, rather than this section which should only include a thorough explanation of the biological and abiotic processes by which grains and oilseeds are contaminated by mycotoxins.
Response: We are sorry for these errors, and we have corrected the information in the original manuscript that "Aflatoxin B1 is classified as a Class I carcinogen by the International Agency for Research on Cancer" repeated in lines 40-43 and 113-114 to lines 44-58. In addition, for the various types of grains and oilseeds repeatedly mentioned in the original manuscript (lines 35-36, 85-86, 117-118, 338-339), the repeated parts were modified or deleted, and for the 35-36 lines in the original manuscript, we chose to retain; We have changed lines 85-86 of the original manuscript, please see lines 119-121; The duplicated parts of lines 117-118 and 338-339 in the original manuscript were deleted by us. The historical information on disease X in Turkey that appeared in the original manuscript (lines 86-88) is moved to lines 43-44 in the introduction.
Comments #7: It is also suggested to ensure that each section contains all the complete information on the subject. For example, Table 1 on regulations on various mycotoxins omits regulations on aflatoxins, which contradicts several statements in the manuscript that point to these mycotoxins as the most common and dangerous, in addition to the fact that these regulations exist in different countries. In the same way, Figure 2 deduces from climate change absolutely all the environmental impact on contamination by these toxins, when environmental factors persistently act and influence contamination with and without this climate change. Likewise, in Figure 3 the absence of post-harvest control measures is notorious, which also contradicts several statements in the manuscript that point to fungi as dangerous in the post-harvest stage and storage of grains and oilseeds.
Response: Firstly, we have added the limitations of aflatoxin B1 in China and the European Union (EU) and the limitations of aflatoxin B1, B2, G1 and G2 in the EU in Table 1; then, we have modified "Climate Change" to "Environmental Factors" in Fig. 2 because environmental factors continue to have an effect on mycotoxin contamination regardless of whether the climate changes or not; and finally, we supplemented Fig. 3 with the mycotoxin control measures for quick drying of grain and oil crops after harvesting.
Comments #8: The manuscript contains several inappropriate styles that we suggest correcting to improve the readability and comprehension of the manuscript. For example, the abbreviation for "aflatoxins (AFs)" is identified (line 39), but later these same mycotoxins are referred to as "AF" (lines 39, 41, 89, 112, etc.). Table 1 contains multiple abbreviations that are not defined in the table itself so one must resort to the text. The provisional maximum tolerable daily intake is defined in two ways (PTMDI and PMTDI) (lines 50 and 60). In the paragraph of lines 62-66 the simile of the Sword of Damocles is made with the contamination of grains and oilseeds by mycotoxins; which is an expression of vague meaning because each reader can interpret it differently according to his knowledge of classical Greek literature.
Response: We apologize for these errors. Firstly, we have standardized the abbreviation of aflatoxins, which appears frequently in the text, to "AFs". Secondly, we have corrected the abbreviation of the Provisional Maximum Tolerated Daily Intake to PMTDI (lines 476 and 486). Finally, the metaphor of the "sword of Damocles" borrowed from the original text (lines 62-66) has been revised to read: "Mycotoxin contamination is a major threat to the quality of grain and oil crop products, and therefore the prevention and control of mycotoxin contamination and grain and oil crops is an important task for all major grain and oil producing countries". Please see lines 62-65.

Reviewer 2 Report
Comments and Suggestions for Authors
Dear Authors
The present review article entitled “The Mycotoxin Contamination in Grain and Oil Crops” discuss about mycotoxins and their concern for environmental and human health. There are certain opportunities for further improvement, please find them below.
1. First, the English language needs revision, some sentences may be reorganized. For example, line 13-17, “In this paper, we summarized the harmful effects of main mycotoxins of grain and oil crops. Then, we analyzed the environmental factors that impact mycotoxin contamination. Finally, we focused control measures of mycotoxin contamination especially the biological control strategies. It is of great significance in the control of mycotoxin contamination in grain and oil crops.” This text may be written as “the current review aimed to summarized the harmful effects of major mycotoxins of grain and oil crops and the environmental factors that impact mycotoxin contamination. Further, control measures of mycotoxin contamination especially the biological control strategies were discussed.
2. Table 1 can be moved to a separate section as regulatory mechanisms of mycotoxins. Introduction may contain the information about mycotoxins, the focused crops of review and its impact on environmental and human health.
3. “Mycotoxins are secondary metabolites produced by fungi” is repeating many times throughout the text, its very general statement, please avoid to include many times.
4. Line 139-157, Please provide the proper references for all the claims made.
5. The review article looks like a compilation of reference list of previous work done in mycotoxins. Highlight the new points of view in the present article and draw a better conclusion. It may also be useful to include a future perspective.
Thank you
Regards
Comments on the Quality of English LanguageThe manuscript needs language revision by a native speaker.
Author Response
Point-by-Point Responses to comments
Reviewer 2:
The present review article entitled “The Mycotoxin Contamination in Grain and Oil Crops” discuss about mycotoxins and their concern for environmental and human health. There are certain opportunities for further improvement, please find them below.
Comments #1: First, the English language needs revision, some sentences may be reorganized. For example, line 13-17, “In this paper, we summarized the harmful effects of main mycotoxins of grain and oil crops. Then, we analyzed the environmental factors that impact mycotoxin contamination. Finally, we focused control measures of mycotoxin contamination especially the biological control strategies. It is of great significance in the control of mycotoxin contamination in grain and oil crops.” This text may be written as “the current review aimed to summarized the harmful effects of major mycotoxins of grain and oil crops and the environmental factors that impact mycotoxin contamination. Further, control measures of mycotoxin contamination especially the biological control strategies were discussed.
Response: We have revised the English language issues that arose in the abstract section. Please see lines 14-18.
Comments #2: Table 1 can be moved to a separate section as regulatory mechanisms of mycotoxins. Introduction may contain the information about mycotoxins, the focused crops of review and its impact on environmental and human health.
Response: We have moved the introductory part of the Table 1 series to the new heading "Regulatory Mechanisms of Mycotoxin" in the section " The Control Strategies for Mycotoxin Contamination in Grain and Oil Crops ". Please see lines 466-490.
Comments #3: “Mycotoxins are secondary metabolites produced by fungi” is repeating many times throughout the text, its very general statement, please avoid to include many times.
Response: We have deleted "Mycotoxins are secondary metabolites produced by fungi in the process of growth and reproduction" from lines 114-116 of the original manuscript.
Comments #4: Line 139-157, Please provide the proper references for all the claims made.
Response: We apologize for those mistakes. We have inserted two references for this paragraph. Please see lines 255-265.
Comments #5: The review article looks like a compilation of reference list of previous work done in mycotoxins. Highlight the new points of view in the present article and draw a better conclusion. It may also be useful to include a future perspective.
Response: We conclude with a new outlook on the use of functional microorganisms and synthetic microbiomes to control mycotoxin contamination of food and oil crops. Please see lines 518-531.

Round 2
Reviewer 1 Report
Comments and Suggestions for Authors
The paper is a review of mycotoxin contamination of grains, as well as the factors that facilitate contamination and the control strategies that are currently employed or have been proposed in various studies. The topic is important from a variety of economic, health and scientific perspectives. The document has been significantly improved by the authors. It is therefore suggested to accept it in its present form.